# Quad-Band Rectenna for Ambient Radio Frequency (RF) Energy Harvesting

**DOI:** 10.3390/s21237838

**Published:** 2021-11-25

**Authors:** Sunanda Roy, Jun Jiat Tiang, Mardeni Bin Roslee, Md Tanvir Ahmed, Abbas Z. Kouzani, M. A. Parvez Mahmud

**Affiliations:** 1Faculty of Engineering, Multimedia University, Persiaran Multimedia, Cyberjaya 63000, Malaysia; jmyjtiang@mmu.edu.my (J.J.T.); mardeni.roslee@mmu.edu.my (M.B.R.); tanvir1533@gmail.com (M.T.A.); 2School of Engineering, Deakin University, Waurn Ponds, Geelong, VIC 3216, Australia; abbas.kouzani@deakin.edu.au (A.Z.K.); m.a.mahmud@deakin.edu.au (M.A.P.M.)

**Keywords:** log-periodic antenna, quad-band rectifier, RF energy harvesting, low power sensor, ambient environment, IMN

## Abstract

RF power is broadly available in both urban and semi-urban areas and thus exhibits as a promising candidate for ambient energy scavenging sources. In this research, a high-efficiency quad-band rectenna is designed for ambient RF wireless energy scavenging over the frequency range from 0.8 to 2.5 GHz. Firstly, the detailed characteristics (i.e., available frequency bands and associated power density levels) of the ambient RF power are studied and analyzed. The data (i.e., RF survey results) are then applied to aid the design of a new quad-band RF harvester. A newly designed impedance matching network (IMN) with an additional L-network in a third-branch of dual-port rectifier circuit is familiarized to increase the performance and RF-to-DC conversion efficiency of the harvester with comparatively very low input RF power density levels. A dual-polarized multi-frequency bow-tie antenna is designed, which has a wide bandwidth (BW) and is miniature in size. The dual cross planer structure internal triangular shape and co-axial feeding are used to decrease the size and enhance the antenna performance. Consequently, the suggested RF harvester is designed to cover all available frequency bands, including part of most mobile phone and wireless local area network (WLAN) bands in Malaysia, while the optimum resistance value for maximum dc rectification efficiency (up to 48%) is from 1 to 10 kΩ. The measurement result in the ambient environment (i.e., both indoor and outdoor) depicts that the new harvester is able to harvest dc voltage of 124.3 and 191.0 mV, respectively, which can be used for low power sensors and wireless applications.

## 1. Introduction

Researchers have tried to investigate various renewable energy sources for different applications over the years. The ambient wireless power density is growing with the exponential and rapid growth of wireless technology, as there is an increasing number of different electromagnetic power sources, such as cellular mobile base stations, digital television (TV) towers, and Wi-Fi routers. In recent years, the concept of using radiofrequency as a power source for low-duty-cycle electronic devices has gained a great deal of popularity to replace the battery and save maintenance costs. Wireless energy harvesting is a feasible approach to transform the ambient RF power to usable dc power by using rectifying antenna (rectenna) technologies. Therefore, the study of the rectenna, which is the most commonly used system for wireless power transmission (WPT) and energy harvesting over the past ten years or so, has made a lot of progress. 

In the past, different forms of a single band, multi-band, and broad-band rectenna have been suggested [1,2,3,4]. A planar rectenna for the ISM band (2.45 GHz) with a high gain of 11.5 dBi has been proposed in [1], where it has been shown that the proposed rectenna can be used to harvest energy to turn on the light-emitting diode (LED) from a distance of 2.8 m. A high-efficiency rectenna was presented in [2] for harnessing energy in the 2.45 GHz frequency band. The rectenna is produced on a relatively expensive RT/Duroid 6002 substrate with a dielectric constant of 2.94 and a loss tangent of 0.0012, while the gain of the proposed antenna in [2] is very high (8.6 dBi). A 2.45 GHz rectenna based on a dual linear square aperture-coupled patch antenna was recently suggested in [3], where a cross-shaped slot is etched on the patch surface, resulting in a 32.5 percent reduction in patch size. For this setup, however, the efficiency obtained is just 38.2 percent. In [4], a broad-band dual-polarized cross dipole antenna with the property of harmonic rejection was used.

A planar dual-band monopole antenna was presented in [5] for energy harvesting in the Global System for Mobile Communications (GSM) bands. However, in their case, the peak gains are very low (1.97 and 3.05 dBi) for the two operating bands. A dual-band rectenna operating in the GSM 1800 and UMTS 2100 bands has been proposed in [6]. Due to the use of an array configuration instead of a single element, the antenna has a sufficiently high gain of 10.9 and 13.3 dBi at 1.85 and 2.15 GHz, respectively. This makes the structure, however, less acceptable for applications requiring a reduced footprint of the antenna. A dual-band rectenna operating at 2.45 and 5.8 GHz has been proposed with integrated novel Combined Performance Superior (CPS) filters and printed dipole antenna in [7]. A triple-band antenna with four rectifier circuit stages operating at 940 MHz, 1.95 GHz, and 2.44 GHz with a realized antenna gain of 0.3, 2.3, and 3.5 dBi was suggested in [8]. A review study was conducted in [9] in which different rectennas, including frequency and power rectennas, were addressed. With 5.2 dBi gain, a wideband twin-loop antenna has been developed. Nine percent RF-to-DC conversion efficiency has been recorded for −20 dBm input RF power [10]. Most of the rectennas discussed above, however, employ the traditional single input configuration at specified frequency points with moderate gain.

Different models have already been investigated, such as single-band rectennas and arrays [11,12,13,14,15,16,17,18], multi-band rectennas [6,19,20,21], and broad-band rectenna arrays [22]; and various other forms of antennas and rectifier designs have also been analyzed and summarised in [23,24]. The overall performance of a rectenna is usually determined by the antenna performance and the rectification circuit’s conversion efficiency. A single narrow-band model is conducive to high performance, but the output power of dc is reduced. A multi-band or a broad-band design or a rectenna array will accumulate more power from weak ambient sources and generate more power output than a narrow-band rectenna, but the trade-offs may be a reduction in overall performance and an increased dimension. The critical problem of wireless energy harvesting at present is how to increase the performance of power conversion over a large frequency band at low-input power levels. Some methods have been used to improve antenna efficiency, such as polarization diversity [25,26,27]. Using a filter between the antenna and the rectifier to reject the higher-order harmonics produced by the nonlinear rectifying circuit [28], the power conversion efficiency can be improved. Using an antenna filter structure, some designs, such as [29,30], have embedded a harmonic-rejection property on the receiving antenna to replace the filter. Besides, the potential of using an adaptive rectifier to balance the complex input power level is addressed in [31] and to control the volatility of ambient incident signals in [32].

A field measurement survey has been performed by some researchers to determine the frequency band and the power density of ambient wireless energy [33,34,35]. The frequency band measured was relatively wide (from 500 MHz to 3 GHz), and the difference in power levels recorded was very important. We have also conducted a field measurement study at Multimedia University in Cyberjaya, a sub-urban region in Malaysia, to gain a better understanding of ambient wireless energy. The cellular mobile radio and WLAN bands of 800–960 MHz, 1790–1880 MHz, 2100–2170 MHz, and 2380–2450 MHz are the frequency bands of interest. There are three types in the measurement area: (1) indoor scenario, (2) outdoor scenario, and (3) semi-indoor scenario. In the outdoor scenario, the highest average power density was found to be −7 dBm/m^2^ on the universal mobile telecommunication system (UMTS)-2100 band. In most cases, the average power density ranges between −35 and −10 dBm/m^2^. This paper presents detailed findings.

The majority of recorded rectennas are not optimized for ambient signal levels. For most of these projects, the desired input power levels are much greater than the levels of ambient input power. In this paper, we propose a novel RF energy harvesting multi-band rectenna that works well from 1.8 to 2.5 GHz. As we noticed in our measurement campaign, the rectenna is built and optimized for relatively low input powers (−35 to −10 dBm). In terms of the incident power level as well as the bandwidth, this design is very different from the traditional rectenna design. The power sensitivity is improved by a new rectifier circuit aimed at reducing the consumption of RF power. In addition, a newly designed dual-band impedance matching network with an additional L-network in the third-branch rectifier circuit is intended to enhance the harvester’s output and RF-to-DC conversion efficiency at relatively very low input levels of RF power density. A dual-polarized multi-frequency bow-tie antenna with a large bandwidth (BW) and a miniature size has been developed. The internal triangular shape and co-axial feeding of the dual cross planer structure are used to decrease the size and increase antenna performance. The rectenna system is then manufactured and tested. The measured results show that the rectenna has good sensitivity at low input power levels. Considering a similar amount of input power, our measured dc output power is higher than the other results reported. The proposed RF harvester is therefore designed to cover all frequency bands available, including most of Malaysia’s cell phone and wireless local area network bands, while the optimum resistance value for full dc rectification efficiency (up to 48%) is from 1 kΩ to 10 kΩ. The outcome of the calculation in the ambient setting (i.e., both indoor and outdoor) shows that the prototype can collect 124.3 and 191.0 mV dc voltage, respectively, which can be used for sensors and wireless applications with low power. 

The rest of this paper is organized as follows. Section 2 explains the configuration of the proposed dual-polarized multi-frequency bow-tie antenna that includes the design of a dipole shape inside a circle with an inner equilateral triangular shape and its performances. Section 3 describes the dual-port quad-band rectifier configuration and performance. Section 4 describes the experimental results of the rectenna in the indoor and outdoor ambient environment. Finally, a conclusion is drawn in Section 5.

## 2. Dual-Polarized Multi-Frequency Bow-Tie Antenna Structure

A self-complementary bow-tie cross dipoles with log-periodic characteristic multi-band antenna is proposed as a receiving unit due to its high bandwidth, omnidirectional radiation pattern, high gain, and multi-beam characteristics. The antenna is made on a 1.6 mm thick FR4 substrate material with relative permittivity of 5.4 and a loss tangent of 0.02. The dimension of the used substrate material is a length of 160 mm (0.29$\lambda_{0}$) and a width of 160 mm (0.29$\lambda_{0}$) at 0.55$\lambda_{0}$ GHz. The 69.00 mm and 80^0^ are the radius and the angle respectively of the selected dipole structure for both front and back layers. In Figure 1, the dual pairs of dipole shapes are formed on both sides of the substrate and are perpendicular to each other. The top layer of the proposed antenna is fed by the inner conductor of the 50 Ω co-axial cable, while the bottom layer is connected by the outer conductor. The fabricated prototype image of the proposed bow-tie log-periodic characteristic antenna is shown in Figure 2. The simulated scattering parameter (S_11_) of the proposed bow-tie cross dipoles multi-band antenna is shown in Figure 3. It can be observed that the suggested multi-band bow-tie antenna (Figure 1a) resonates at 1 GHz with a BW of 258 MHz, but the performance of the impedance matching is less than −7 dB. In order to get higher impedance BW within the range of frequency at 800 MHz to 2.7 GHz, the self-complementary bow-tie vacant dipole structure is updated by an inner triangle structure inside a circular shape, as depicted in Figure 1b.

A circle is inscribed in the triangle where the triangle’s three sides are all tangents to the circle. As the triangle’s three sides are all tangents to the inscribed circle, the distances from the circle’s center are equal to the radius. The radius *r* of the inner circle is calculated in such a way that it is equal to the height of the triangle. The unequal length of the arms *a*, *b*, and *c* of both front and back-sided triangles is shown in Figure 2a. The minimal resonate frequency of the proposed multi-band bow-tie antenna is identified by the length of the dipole, which is equal to λ_0_/4 in length. In Figure 1c, the minimum resonant frequency of the self-complementary dipoles (length 69.00 mm) is nearby 1 GHz, while the resonance frequencies cover between 800 MHz and 2.7 GHz. But due to impedance mismatch, the available frequency bands within the range of 0.5 to 3 GHz are not covered as well as the average level of simulated return S11 loss is about −7.25 dB for the fifth resonance band. In order to increase the value of the scattering parameter of the proposed bow-tie antenna, the diagonal set of dipoles are modified with a triangle inside a circular shape which is connected to a novel co-axial feeding that can produce the dual circular polarized radiation field to progress the performance parameters of self-complementary new dipoles structure.

The dual pairs of the cross dipole are connected by a 50 Ω co-axial feeding in the center of the antenna to generate a 90-degree phase delay and form the right-hand circular polarization radiation field and left-hand circular polarization radiation field in front and backside of the antenna, respectively in Figure 1d. It is noticeable that impedance matching can be achieved by using the co-axial feeding technique. The average signal level of the simulated scattering parameter (S11) beyond the bandwidth has been developed from −9.2 dB to −20 dB. Moreover, a triangle with 2.5 mm wide arms is designed inside the dipoles on the substrate to cover the frequency bands lower than 1 GHz. The single complementary sets of modified dipole structures are situated in a diagonal position of the proposed antenna. The measured S-parameter of the proposed antenna is showed in Figure 3. It has been shown that the comparison between simulated and measured s-parameters of the proposed antenna is good in agreement. A circular shape is designed inside the triangle on both the front and backside of the printed circuit board (PCB) board to occupy the last target frequency band. In order to fulfill the condition for circularly polarized radiation pattern (Figure 4) and improve the impedance matching of the bow-tie multi-band antenna, the single complementary pair of the vacant dipole in a single diagonal position and another single pair of modified dipole structure in other diagonal position is placed both upper and bottom layer of the substrate. Figure 5 shows the simulated and measured evaluation of the realized gains along with the frequency band of the proposed bow-tie antenna. The measured realized gain has achieved a higher gain than the simulated gain in the band of 1.83, 2.19, and 2.45 GHz, except for the band of 0.89 GHz.

There are many factors that affect the antenna performances and showed discrepancies between simulations and measurements performances. The most common factors are the fidelity of simulation (esp. accuracy of model and geometry and materials attributes) and fidelity of measurement (calibration, outside influences and interferers, etc.). Maximum and minimum measured gains are reached at 5 and 3.7 dBi, respectively.

The surface current distribution of the antennas are shown in Figure 6, which provides a clear understanding of the proposed bow-tie antenna’s behavior by demonstrating the current distributions along the vacant diploe with inner triangle and circular structure at 0.89, 1.83, 2.19, and 2.45 GHz, respectively. It is observed that the current flows through the dipole structure, but the majority of current particles exist near the joining point of each dipole. Surprisingly, a higher density of current particles flows through the arms of triangle shapes when the frequency level is below 1 GHz, which validates the characteristic of self-complementary in the proposed antenna. The simulated and measured 2D radiation patterns of E and H-field of the proposed bow-tie antenna at 0.89, 1.83, 2.19, and 2.45 GHz are illustrated in corresponding figure. The antenna covers the preferred frequency bands, and it has the broadside directional polarization features for the majority of the resonator bands (except 0.89 GHz), but it has both a stable dipolar pattern on the yoz-plane (E-plane) and a stable omnidirectional pattern on the xoz-plane (H-plane). It can be seen that a better front-to-back ratio is obtained at the higher resonance frequency bands. The radiation pattern was gradually distorted with increasing frequency and propagation distance. Observed features suggest that propagating wave scattering due to small-scale velocity heterogeneity in the crust may be a major cause of this distortion. The effects of propagating wave scattering on apparent primary wave radiation pattern were investigated via 3-D finite difference simulation of EM wave propagation. The simulations are demonstrated that the scattering of EM waves modified the apparent primary wave radiation pattern from the original four-lobe shape, and that the small-scale velocity heterogeneity, characterized by the von Kármán-type power spectral density function. It was also found that the scattering attenuation of primary wave expected from this heterogeneity is significantly smaller than the apparent primary wave attenuation and *S*-wave scattering attenuation. The isotropic pattern is achieved through an excellent choice of the triangle and circular shapes for the antenna; the maximum directivity is slightly deviating from *x* and *y*-axis with increasing frequency ranges (i.e., 3.72, 4.59, 4.56, and 4.86 dBi at 0.89, 1.83, 2.19, and 2.45 GHz, respectively).

## 3. Dual Port Quad-band Rectifier Design

In order to achieve good balancing between antenna and rectifier and mitigate the circuit complexity, a dual-port quad-band rectifier is designed to harvest the RF energy with low power RF density level from the ambient environment. In Figure 7, the second rectifier is a modified half-wave Greinacher rectifying circuit, and the first rectifier (Figure 8) is a conventional voltage doubler rectifying circuit. With the combination of rectifier circuits 1 and 2, the proposed dual ports rectifier circuit is able to cover the available frequency bands (i.e., GSM 900, GSM 1800, 3G, and Wi-Fi). By optimizing every branch of the rectifier 1 and 2, the RF-to-DC rectification efficiency is better tuned in receiving a local maximum for every sub-frequency band of interest. The topology of the suggested rectifier is depicted in Figure 9. The performance parameters of the dual-port rectifier are optimized to attain the maximum RF-to-DC rectification efficiency for all existing frequency bands with associated RF input power density levels of −30 to −20 dBm. Advanced Design System (ADS) v19 is used to design the quad-band rectifier. The connection of two antennas with two ports of the proposed rectifier is not a promising technique for achieving better efficiency because of its relatively small impedance bandwidth and impractical configuration. Due to the frequency-dependent input impedance of the Schottky diode and the narrow BW performance of common impedance matching networks, it is difficult to achieve large-signal input matching overall available frequency bands, which is crucial to assure the maximum power absorbed by the load. From the literature review of [8,19,35,36]), single branch multi-band rectifiers are not able to guarantee the coverage of the available, expected frequency bands while allowing for very low ambient RF power operation. To achieve a high RF-to-dc rectification efficiency of the rectifier over all the frequency bands, two single series diodes have been selected to connect with impedance matching network in parallel, in such a way that each of them operates in a wideband [37], effectively adjusting the overall frequency band. The scattering parameter files supplied for the surface mount device (SMD) inductors by the coilcraft 1080HP series manufacturer are introduced in the ADS schematic to ensure the accurate optimization process.

The low-barrier Schottky diode of HSMS2850 and low biasing SMS7630 diodes are used, suitable for operation in the μW range [38]. Every Schottky diode is series connected with IMN that are consisting of multi stubs microstrip matching elements (i.e., open stubs, a short stub, a radial stub, a meander line, and a taper), aimed at adapting the 50 Ω input impedance of the antenna to the complex conjugate of the large-signal input impedance of the selected diode (see [39]). IMN is designed to transfer the maximum power from the source to the load. The IMN can increase the voltage gain before the RF rectifier to dominate the threshold voltage as a result of increasing the RF to DC rectification efficiency. At the specific operating frequency range, the impedance between source and load is matched so that impedances are complex conjugate to each other. There are two ways to design IMN: using lumped components (i.e., inductor and capacitor) and distributed microstrip components (i.e., open stub, short stub, and mender line). In this research, the proposed rectifier consists of a voltage doubler and a half-wave Greinacher rectifier. Here, microstrip matching elements such as radial stub, short stub, and mender line are used to design the IMN because they can transfer maximum power from the source to load. Moreover, the quality factor (Q-factor) of such a type of matching network boosts the threshold voltage level and offers the passive and strong amplification of the RF input signal. 

In rectifier 1, the IMN (radial stub and short) is designed in such a way that it covers GSM 900. Similarly, in rectifier 2 the rectifying microstrip matching elements such as radial stub, short stub and, open stub, meander line are used to design IMN for both the first and second branch so that combinedly they can cover GSM 1800, 3G, and Wi-Fi frequency bands. In this design, the resulting IMN are quad-band MN with better Q-factor concerning wideband design, which is considered in the development of DC- rectification efficiency. It is observed that the modified rectifier topology (rectifier 1 and 2) is especially highly sensitive to the losses of the transmission line, which connects the selected diode to the stub of each branch so that it drops maximum voltage. 

The simulations are conducted by taking into consideration the high-impedance microstrip transmission lines connected to the ground and the S-parameter files of the inductors supplied by coilcraft. Similarly, the first rectifier circuit (voltage doubler rectifier) consists of a matching network between rectifier sections so that it can operate in a low-frequency band. The proposed design is produced for the −20 dBm input power with input impedance of 0.721 + j0.121 @ 0.850 GHz, 0.916 + j0.008 @ 1.81 GHz, 0.917 + j0.009 @ 2.12 GHz and 0.938 + j0.010 @ 2.41 GHz as shown in Figure 10. The rotation of the input impedance is in such a way around the center of the smith chart so that the resonance frequencies outside the expected operating sub-band characterized a high impedance. On the other hand, the operating frequency bands of the dual-port rectifier are slightly affected by its corresponding branch because the rectifier is designed in such a way that the input impedance at each frequency band is nearly placed at the center (50 Ω) of the Smith chart. 

The prototype image of the dual-port rectifier is shown in Figure 11. The rectifier topology permits each frequency band to enter the dedicated branch which is optimized to transform it, thus declining the subsequent loss of power. Figure 12 illustrates the input reflection coefficient for both ports of the rectifier. These reflection coefficients are calculated for the ambient RF power density level of −20 dBm. The optimized load value of the rectifier is 1.5 kΩ (see Figure 13). Finally, the complete multi-band prototype rectifier is realized with two different types of Schottky diodes (two HMSM 2850 and two SMS 7630), two Coilcraft inductors, seven capacitors and a single load resistor. After a fine-optimizing of the value of the different lumped components, which has been performed to compensate for supplementary parasitic elements not incorporated in the simulations (such as the soldering effects on the prototype and the connectors affect, and so on), the performance of the rectifier based on a single tone sweep has been examined for different RF power density levels, and the results are reported in Figure 12, Figure 13, Figure 14 and Figure 15.

Figure 12 depicts the available RF signal input scattering parameter value at both ports of the proposed rectifier with the variation of ambient input RF power density levels. The reflection coefficient |S11| value of less than −10 dB was achieved across all frequency bands of interest for different RF input powers. It is noticed that the reflection coefficient is shifted from left to right with the decreasing RF input power levels. It also showed a slight difference between simulation and measurement of the reflection coefficient of the proposed dual-port multiband rectifier. The performance of RF-to DC rectification efficiency as a function of different load values is demonstrated in Figure 13 when the input RF power level remains constant at −20 dBm. It is noticeable that the conversion efficiency is greater than 60% at 0.850 GHz, 48% at 1.81 GHz, 35% at 2.18 GHz, and 25% at 2.40 GHz for the load resistor value of 1.5 and 4 kΩ respectively. Comparing the simulated and measured value of conversion efficiency at different frequency bands (i.e., 0.850, 1.81, 2.18, and 2.40 GHz) at −20 dBm RF input power density level shows good agreement.

The achieved maximum RF-to-dc conversion efficiency is about 63% at 0.850 GHz for a load resistance value of 1.5 kΩ, as it is considered as a dual-port rectifier load. The optimum load resistance is different for the triple branches of the rectifier, and the value of 1.5 kΩ is selected due to a trade-off intended at enhancing the dc output voltage under the postulation that an identical amount of generated voltage enters each branch of the rectifier at its corresponding central frequency. Moreover, it is illustrated that the dc rectification efficiency of the quad-band rectifier is maintained for a broad range of selected load resistance which is significant in many practical applications. The RF-to-dc rectification efficiency of the dual-port rectifier can be calculated as
(1)$$\eta_{\mathrm{RF}@input-DC}=\frac{P_{DC}}{P_{\mathrm{RF}@input}}=\frac{V_{DC}\times I_{DC}}{P_{\mathrm{RF}@input}}=\frac{V_{DC}^{2}}{P_{\mathrm{RF}@input}R_{load}}$$
where, $P_{DC}$ is the output dc power, $P_{\mathrm{RF}@input}$ is the input RF power density to the rectifier, $V_{DC}$ is the generated dc output voltage, $I_{DC}$ is the *DC*, and $R_{load}$ is the optimal load resistance. By sweeping the load value from 1 to 7 kΩ throughout the optimization of the rectifier, the optimal value of load resistance is determined to be 1.5 kΩ so that the dc rectification efficiency is maximum. A TSG4104A RF vector signal is used as the source of RF input to the rectifier throughout the measurement. The measured and simulated RF-to-dc rectification efficiency (with the selected load resistance) at the four center frequencies as a function of the input RF power is represented in Figure 14. An excellent agreement between the simulated and measured rectification efficiency is achieved at different frequencies 0.850, 1.81, 2.18, and 2.40 GHz, while the measured rectification efficiency is smaller than the simulated for all frequency bands. This might be because of the conduction loss of the Schottky diodes and the PCB at higher frequency bands and the inaccessible parasitic loss of the SMD components. It is observed that the maximum efficiency is up to 65% for −15 dBm RF input power if the harvester can extract RF power from receiving the signals at the four frequency bands simultaneously. In this circumstance, the total rectification efficiency development is nearly 35%. The RF-to-dc conversion efficiency is increased from 17% (at −20 dBm input) to 48% (at −20 dBm input), which proves that the proposed dual-port rectifier has good power sensitivity and is efficient for the relatively low ambient input power. Figure 15 presents the resulting RF-to-dc rectification efficiency as a function of frequency, which is derived from the Equation (1). An excellent agreement between the simulated and measured rectification efficiency is achieved at different frequencies 0.850, 1.81, 2.18, and 2.40 GHz, while the measured rectification efficiency is smaller than the simulated for all frequency bands. This might be because of the conduction loss of the Schottky diodes and the PCB at higher frequency bands and the inaccessible parasitic loss of the SMD components. It is observed that the maximum efficiency is up to 65% for −15 dBm RF input power if the harvester can extract RF power from receiving the signals at the four frequency bands simultaneously. In this circumstance, the total rectification efficiency development is nearly 35%. The RF-to-dc conversion efficiency is increased from 17% (at −20 dBm input) to 48% (at −20 dBm input), which proves that the proposed dual-port rectifier has good power sensitivity and is efficient for the relatively low ambient input power. Figure 15 presents the resulting RF-to-dc rectification efficiency as a function of frequency, which is derived from Equation (1). The measurement is taken separately from the Port1 (stated in Figure 15a) and for Port 2 (in Figure 15b). The first rectifier shows the respective maximum rectification efficiency of 50% and 42.5% for −15 dBm and −20 dBm RF input, respectively. In the same frequency band, the achieved individual conversion efficiencies are 40% at −30 dBm and 30% at −35 dBm RF input power density levels.

On the other hand, the second rectifier, which characterizes a better performance, shows the peak conversion efficiencies of 65%, 55%, and 19% at −15 dBm RF input power density level (i.e., Figure 15b). Moreover, its conversion efficiency deteriorates to 52%, 45%, and 15% at −20 dBm RF input power level for different frequency bands. The decline of this performance is mainly due to the conduction loss of the diode, which rises with frequency bands [40], and because of the RF input interactions as well as interferences among the three branches of the topology. However, it is noticeable that the rectifier performance fluctuates fairly smoothly in all bands for all RF input power levels under the measurement, thus attaining the goal of harvesting RF energy from available bands of interest within the range. The generated dc voltage is measured by input of various multi-tone signals with signal, dual and triple tones. The measurements were performed using a TSG4104A RF vector signal generator to generate with multitone signal generation capability and an Anritsu MS2024A Master Vector network analyzer (VNA) running on a TTi PSA6005 Spectrum Analyzer as a receiver.

## 4. Measurements of RF Harvester in Ambient Environment (Indoor and Outdoor)

After optimization of both multi-band antenna and rectifier, the multi-band RF harvester was made. There are two phases of measurement procedures that were performed in the following ways:

### 4.1. Employing Artificial RF Energy Sources

The newly designed multi-band RF harvester was connected to a TSG4104A RF vector signal generator to generate a signal for specific frequency bands (i.e., GSM 900, GSM 1800, 3G, and Wi-Fi) with an associated RF power density level of −20 dB (i.e., minimum ambient RF power level) as input. At first, the separate s-parameter response of the dual-port rectifier (i.e., port 1 and port2) was measured by using the Anritsu MS2024A Master Vector network analyzer (VNA) as shown in Figure 16a,b. Secondly, both VNA and a digital multimeter are used to measure the performance of the antenna and rectifier in the applied lab at MMU (i.e., Figure 16c,d).

Here antenna and rectifier are individually wirily connected with Master VNA and signal generator respectively. The prototype of the fabricated multi-band antenna was initially used to exploit the electromagnetic (EM) signals with associated different RF power density levels at a minimum distance that is equivalent to the antenna far-field radiation range. A spectrum analyzer was used to measure and record the received RF power and corresponding transmitting power, respectively. The new harvester was placed in the same location where the antenna was located in the first measurement. The dc output voltage was recorded across the load of the harvester. The transmitting power was varied with respect to the corresponding received RF power density levels that were adjustable from −30 to −5 dBm. The generated dc output power can be determined from the modified Equation (1) by converted in dBm as following
(2)$$P_{dc}\left(\mathrm{dBm}\right)=10log_{10}\left(\frac{V_{DC}^{2}}{R_{load}}\times 10^{3}\right)$$
where, $P_{dc}$ is the output dc power in dBm, $V_{DC}$ is the dc output voltage, and $R_{load}$ is the optimal load resistance. The measured dc output voltage as a function of received RF power density levels for the number of input frequency bands is shown in Figure 17. As shown in Figure 17, the measured output dc voltages are in good agreement with the simulated result. The proposed rectenna does enhance the output dc voltage level as the number of frequency increases. For three consecutive input frequency bands, the measured output dc voltage is greater than the input of single frequency bands. This increase performance is due to an excellent impedance matching between the antenna and port 2 of the rectifier and almost equal to the sum of dc voltage. If some of the incident powers do not have the same dc value, the impedance matching, and hence, the measured output dc voltage is not optimum.

### 4.2. RF Power Source at the Ambient Environment

In order to measure the realistic performance of the proposed harvester, the campus of Multimedia University (i.e., a typical semi-urban environment in Malaysia) is selected where input RF power density level is comparatively low. An isotropic antenna (i.e., 1.8 GHz) is integrated with a TTi PSA6005 (i.e., Figure 18b) spectrum analyzer (i.e., 0 to 6 GHz frequency range) to measure the available frequency bands with associated RF power density levels within the range of 0.5 to 3 GHz. Figure 18a shows the measured ambient frequency bands and corresponding power density levels of the selected location (i.e., within MMU campus). It can be noted that the received power by spectrum analyzer is primarily distributed at four frequency bands which are GSM-900, GSM-1800, 3G, Wi-Fi, and LTE. The input RF power density level over the entire frequency band is around −35 to −24 dBm. The average RF power density level in the band of interest can be determined by using a broad-band power sensor delivered by Rohde and Schwarz [40]. The measured RF power density level from the available frequency bands received by the proposed antenna was fluctuating between −25 and −10 dBm as a function of time. The average ambient RF power level in the available frequency band was nearly about −20 dBm. Finally, the log-periodic antenna was replaced by the RF harvester and the generated dc output voltage was obtained using a digital multimeter, as displayed in Figure 19a, b. The measured dc output voltage of the new harvester is around 120–200 mV in the ambient environment. It can be seen more specifically that the generated dc voltage at port 1 is about 124.3 and 191.0 mV in port 2. This is due to the difference between the number of frequency bands covered by port 1 and port 2. Port1 is responsible for harvesting RF energy from single frequency bands where port 2 simultaneously harvest energy over the three frequency bands in an ambient environment. Using Equation (2), the measured dc power was obtained to be −24 to −20 dBm which appears to be greater than the received RF power levels of −35 to 30 dBm. This is because the proposed harvester is of a wider BW and has amalgamated RF power received from its frequency bands into dc power, and the resultant RF power is, therefore, higher than the RF input power at individual frequency bands. The peak value of overall RF energy rectification efficiency (i.e., about 25.5%) in this situation is obtained by averaging the total received power in the band and then dividing it by the dc output power. In the same scenario, the harvester was also measured multiple times by varying the load resistance. The measured RF-to-dc rectification efficiency as a function of different values of load resistance that varies from 0.5 to 6 kΩ is illustrated in Figure 20 with the error marked by a cross sign.

This is due to the different values of measured dc voltage. It should be noted that the maximum RF-to-dc rectification efficiency was attained with the optimum load resistance of nearly 1.5 kΩ. In Table 1, a comparative study is demonstrated between the proposed harvester and previous related designs. It should be noted that most of the previous research work is for single, dual, and multi-band ambient operations. This design (i.e., dual ports) provides a wideband performance with a higher RF-to-dc rectification efficiency. The compactness of the new design is reasonable than the majority of the previously designed RF harvesters.

It is demonstrated that all those harvesters have been put at the identical location with similar ambient input RF power density levels (i.e., −20 dBm) for all frequency bands (the anticipated measured dc output voltage are given in the last column), the new harvester has generated the maximum dc output voltage because of its high rectification efficiency and broad frequency BW. The majority of the previously published works have not yet been able to meet the sufficient dc output voltage at such a low input RF power density level in an ambient environment.

## 5. Conclusions

The dual ports quad-band RF harvester is suggested using triple branches with a new impedance matching technique. The new design with a matching network for each branch can maintain good performance in various conditions such as multiple bands, different RF input power density levels, and a lower range of load resistance. These characteristics are very crucial for realistic wireless energy scavenging. Both the design technique and the outcomes of the measurement are comprehensively discussed. The new harvester is able to scavenge RF energy in all available ambient frequency bands at input RF power density levels as low as −20 dBm in a wide range of realistic and conformal circumstances, hence setting the base of truly independent IoT and smart node alignments. Further improvements are also possible in terms of multi-tones operation, matching with higher bandpass filer in each branch of first and second rectifiers. The proposed harvester is better than the previous works in terms of the overall rectification efficiency as well as the coverage of available frequency bands and a wide range of load resistance. Considering the excellent performances of the harvester with different ambient conditions, the suggested design is very compatible with many realistic low power sensors and electronic devices and thus can be applied in numerous batteryless wireless applications.

## Figures and Tables

**Figure 1 sensors-21-07838-f001:**
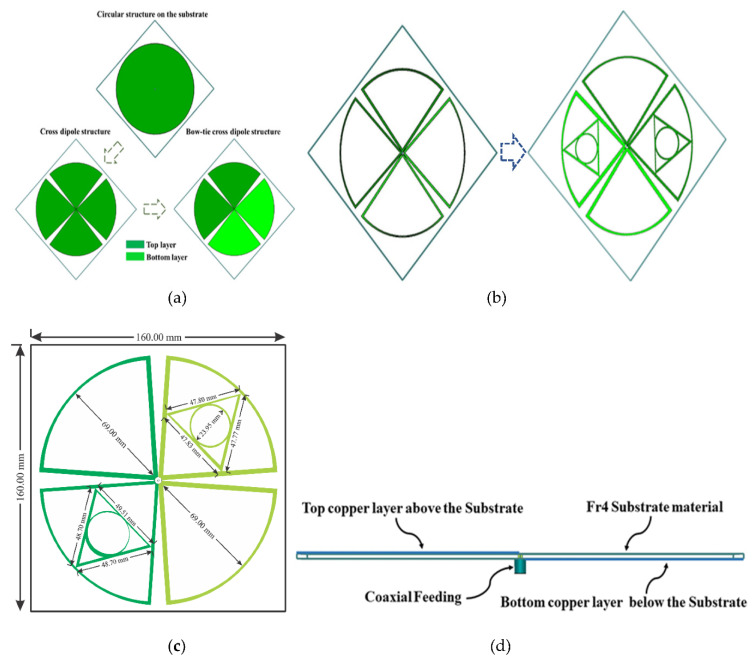
The design process of the proposed antenna; (**a**) conversion of the circular shape into bow-tie cross dipole. (**b**) Regeneration of the dual vacant dipoles inside a triangle shape with inner circular structure (**c**) The optimized dimension of the final design (**d**) a 50 Ω novel co-axial feeding technique with front and backside view.

**Figure 2 sensors-21-07838-f002:**
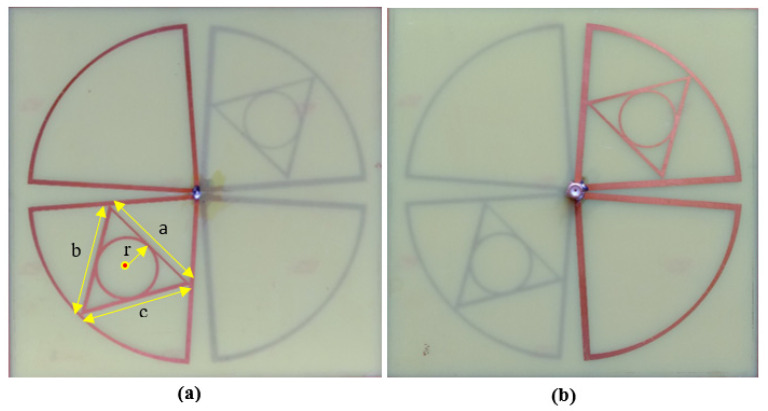
Prototype image of the proposed antenna; (**a**) front and (**b**) backside.

**Figure 3 sensors-21-07838-f003:**
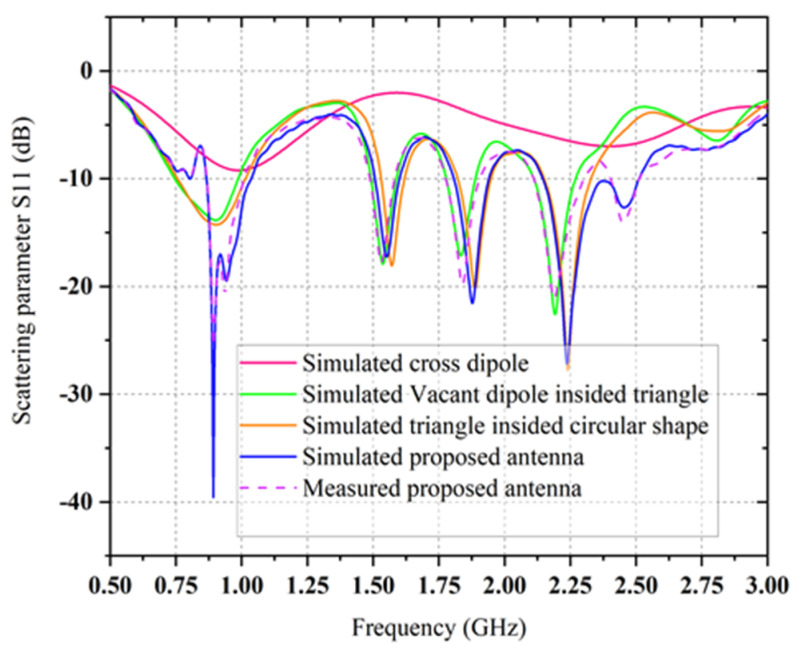
The simulated S-parameter of the two different antennas and the comparison between simulated and measured S-parameter of the proposed bow-tie antenna.

**Figure 4 sensors-21-07838-f004:**
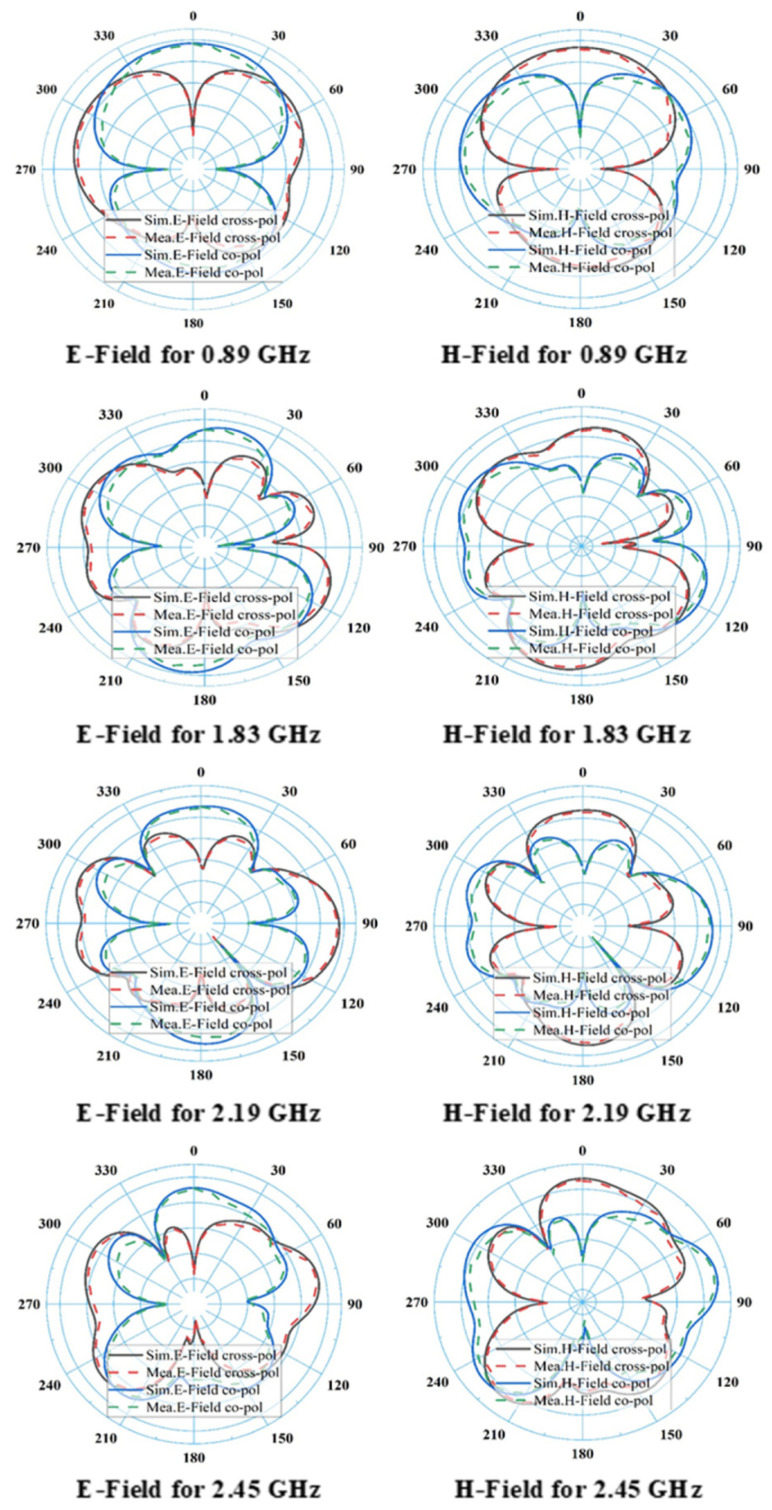
Simulated and measured 2D E and H-field radiation patterns for different frequency bands such as 0.89, 1.83, 2.19, and 2.45 GHz.

**Figure 5 sensors-21-07838-f005:**
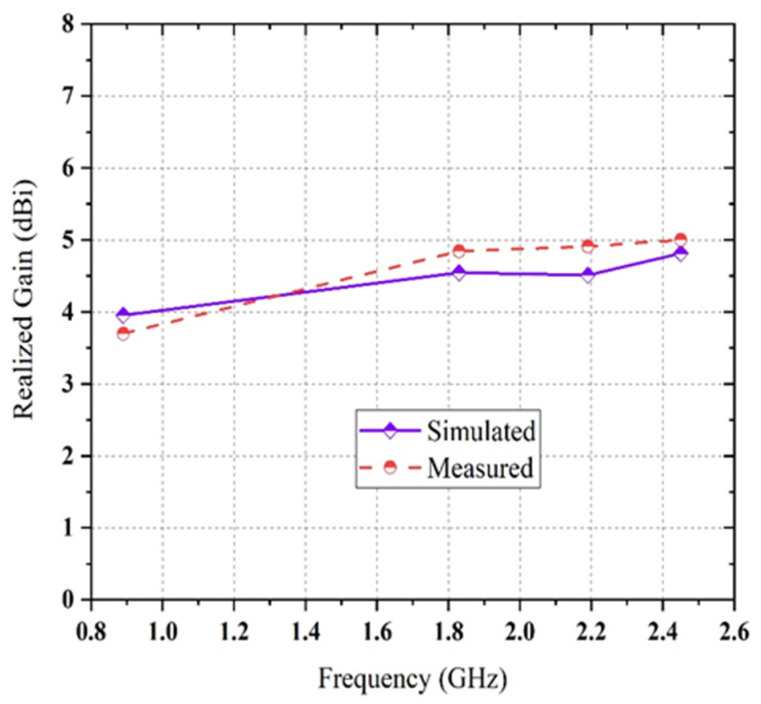
The comparison between simulated and measured realized gains of the proposed antenna.

**Figure 6 sensors-21-07838-f006:**
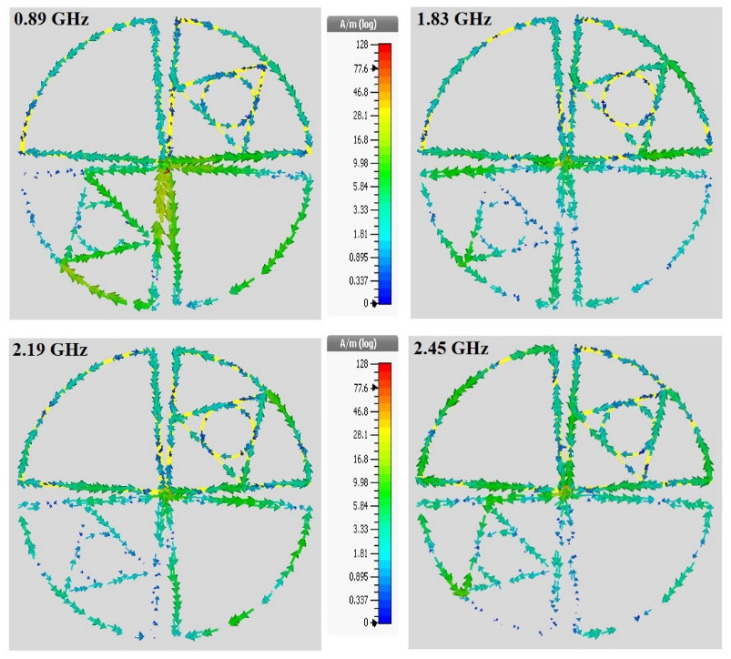
The surface current distribution at the different frequency bands (i.e., 0.89, 1.83, 2.19, and 2.45 GHz.

**Figure 7 sensors-21-07838-f007:**
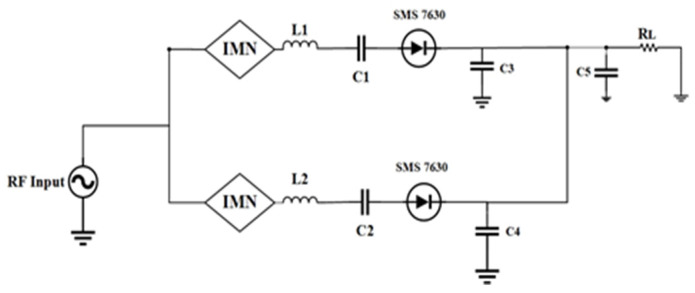
Topology of the novel half-wave Greinacher rectifier circuit with two-branch impedance matching circuit (IMN = impedance matching network).

**Figure 8 sensors-21-07838-f008:**
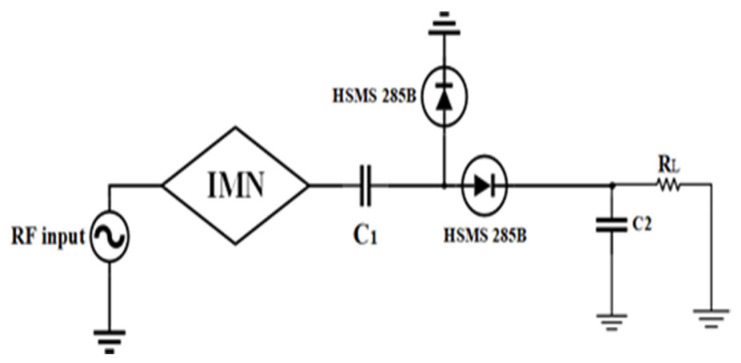
Topology of a conventional voltage doubler rectifier circuit (IMN = impedance matching network).

**Figure 9 sensors-21-07838-f009:**
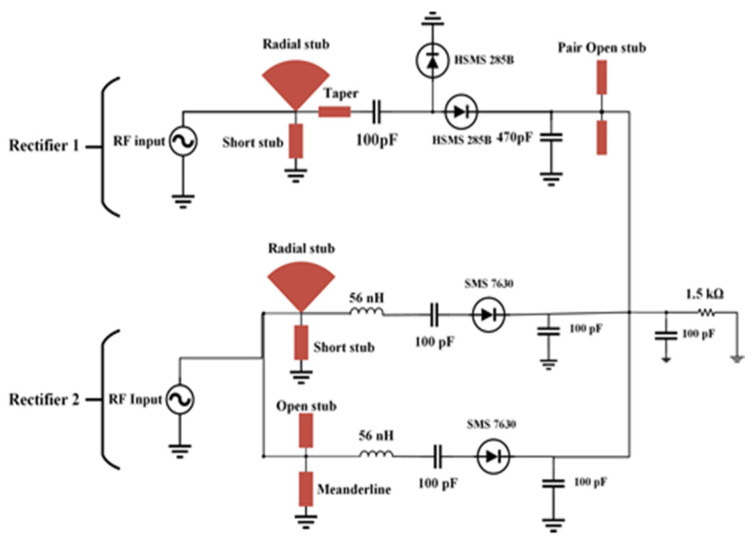
The proposed dual ports quad-band rectifier topology.

**Figure 10 sensors-21-07838-f010:**
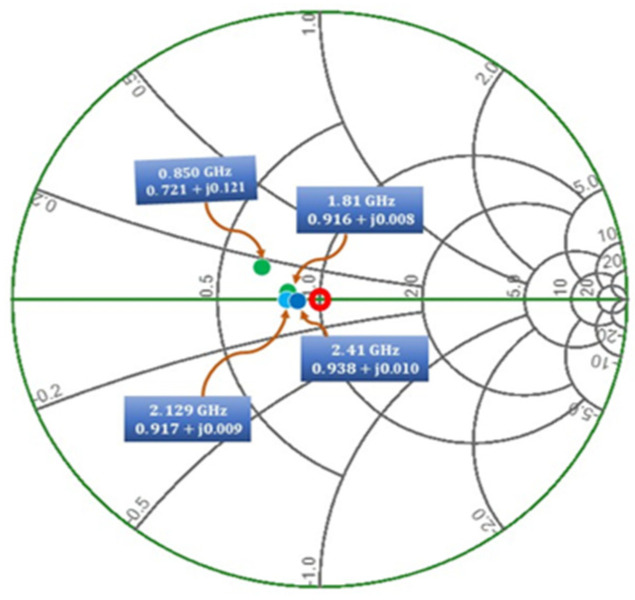
Study of rectifiers 1 and 2: input reflection coefficient of each frequency band corresponding impedance. The simulation results are reported for an available input power of −20 dBm.

**Figure 11 sensors-21-07838-f011:**
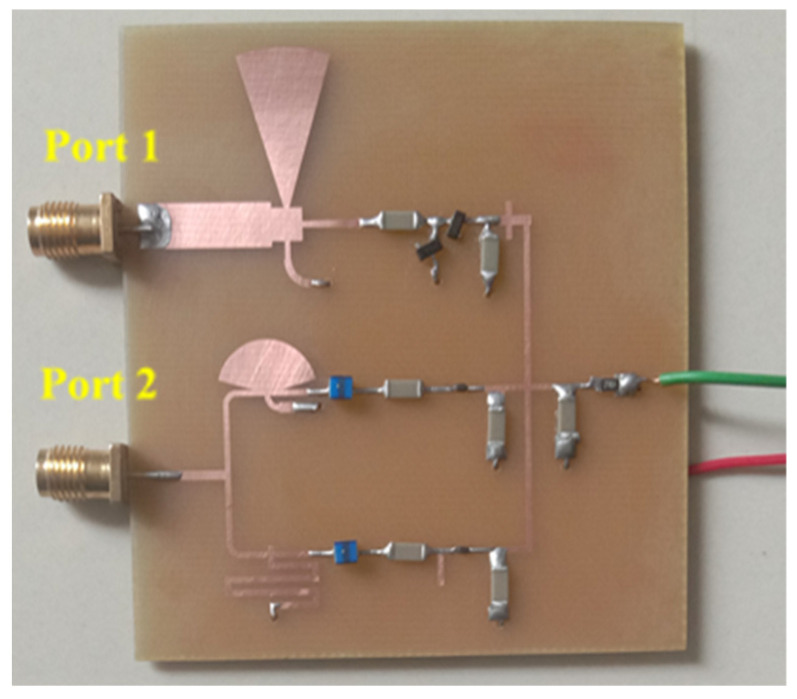
The prototype image of the proposed dual-port rectifier topology.

**Figure 12 sensors-21-07838-f012:**
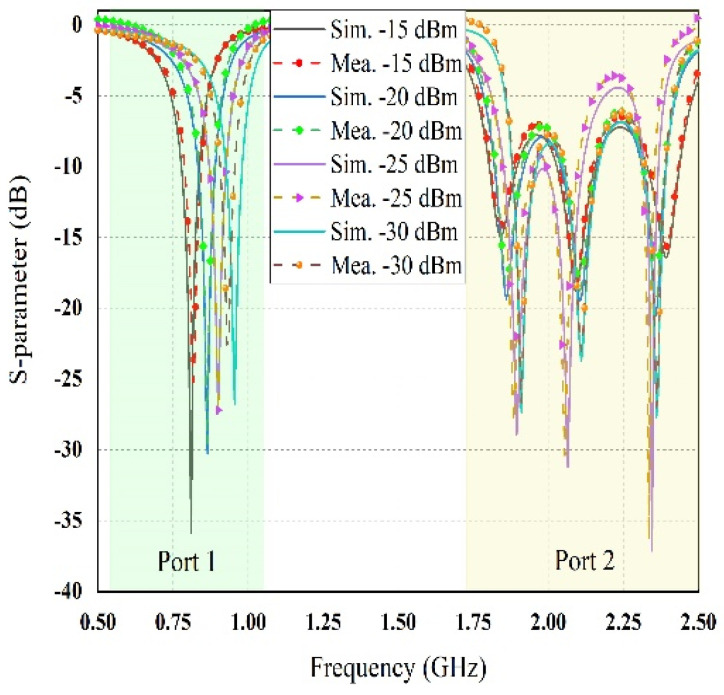
The comparison between simulated and measured reflection coefficient of port 1 and port 2 with the variation of RF power density levels.

**Figure 13 sensors-21-07838-f013:**
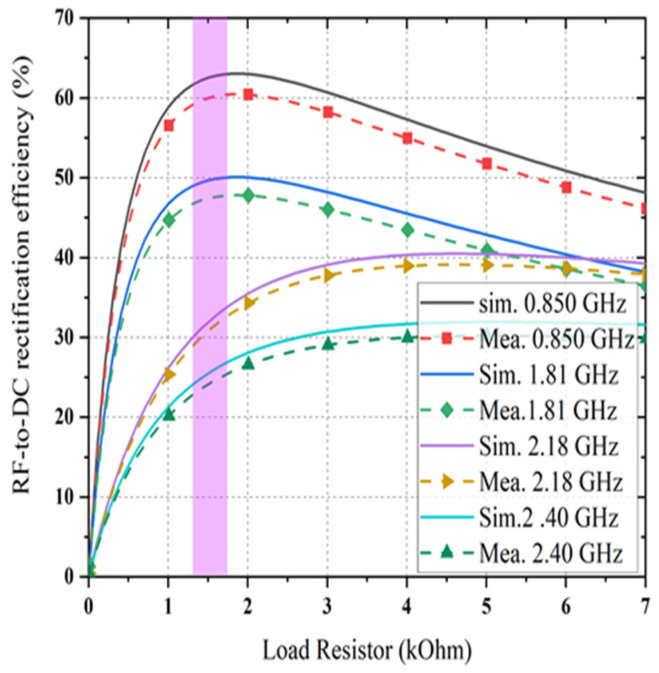
The RF-to-dc rectification efficiency as a function of load resistance when RF input power level of −20 dBm.

**Figure 14 sensors-21-07838-f014:**
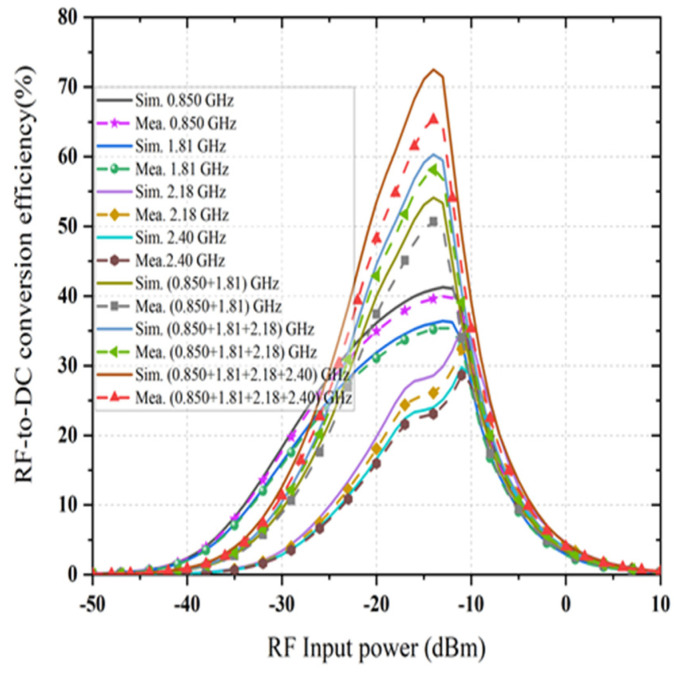
Simulated and measured RF-to-dc conversion efficiency of the rectifier versus input RF power level at four frequency bands. Selected load resistance 1.5 kΩ.

**Figure 15 sensors-21-07838-f015:**
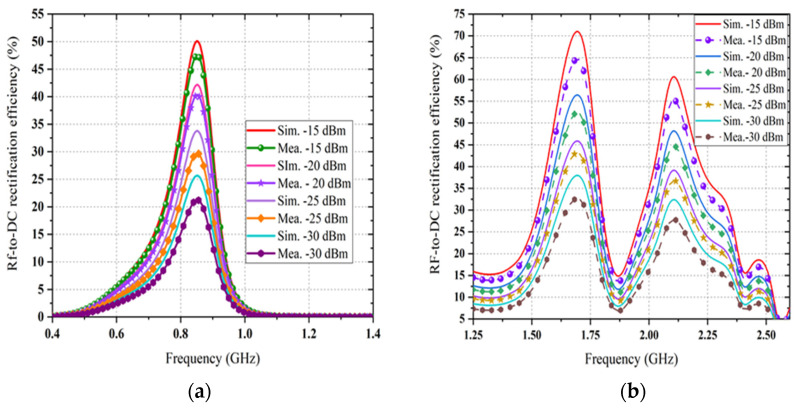
This Simulated and measured RF-to-dc conversion efficiency of the rectifier versus frequency at different input RF power density levels for (**a**) port 1 and (**b**) port 2.

**Figure 16 sensors-21-07838-f016:**
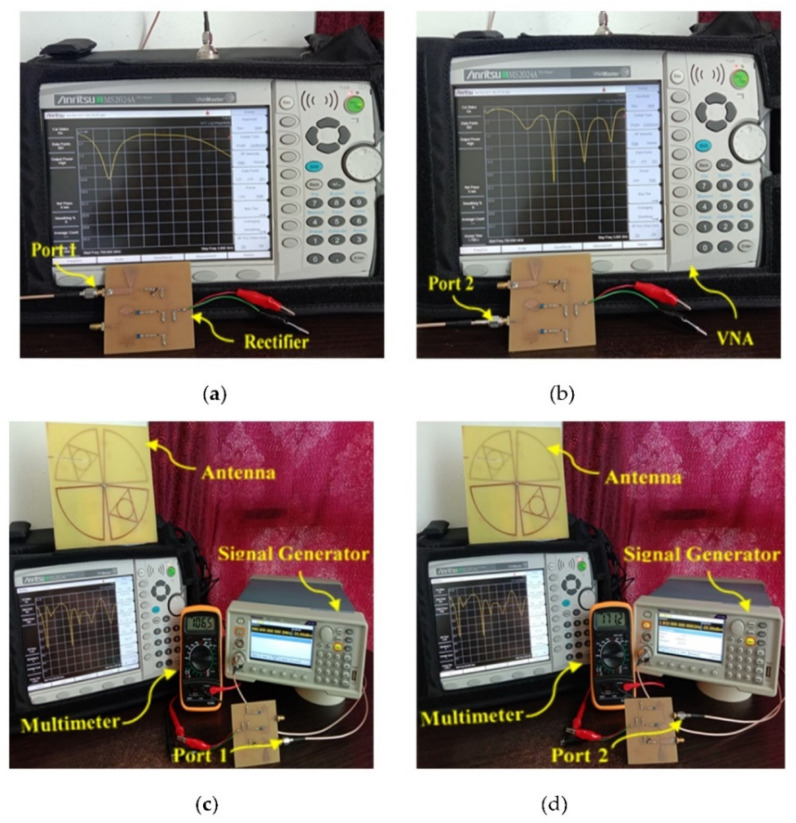
Measured s-parameter of the (**a**) rectifier port1, (**b**) rectifier port 2, (**c**) antenna and generated dc voltage for port 1, and (**d**) antenna and generated dc voltage for port 2.

**Figure 17 sensors-21-07838-f017:**
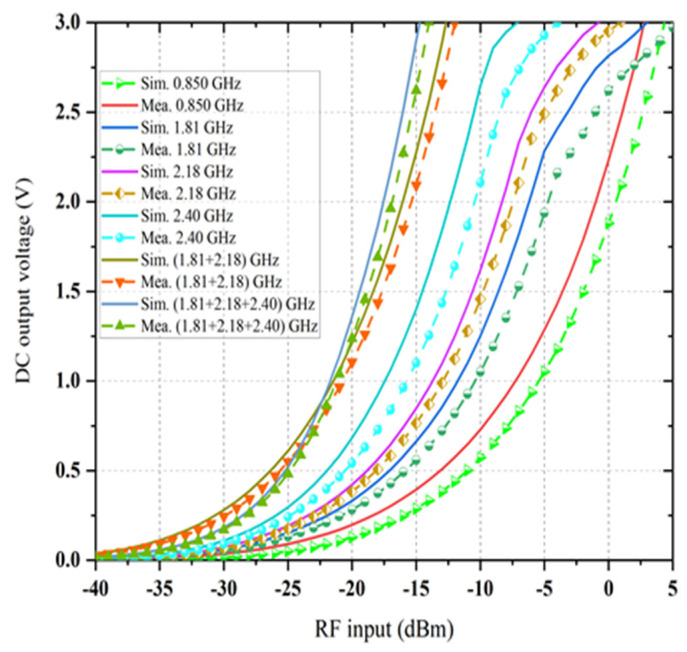
Simulated and measured RF-to-dc conversion efficiency of the rectifier versus frequency at different input RF power density levels.

**Figure 18 sensors-21-07838-f018:**
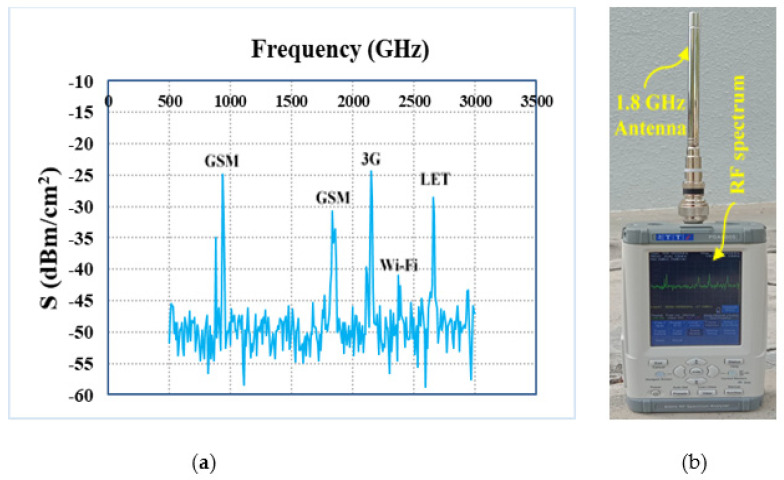
(**a**) The available frequency bands with associated RF power density levels within the area of MMU campus and (**b**) TTi PSA6005 spectrum analyzer with calibrated antenna and complete RF spectrum.

**Figure 19 sensors-21-07838-f019:**
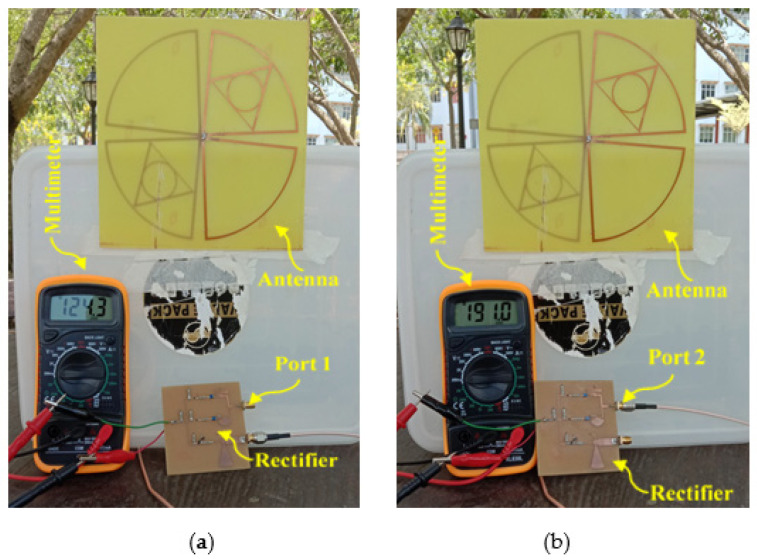
Measured output dc voltage (**a**) port 1 and (**b**) port 2 of the proposed harvester at the ambient environment.

**Figure 20 sensors-21-07838-f020:**
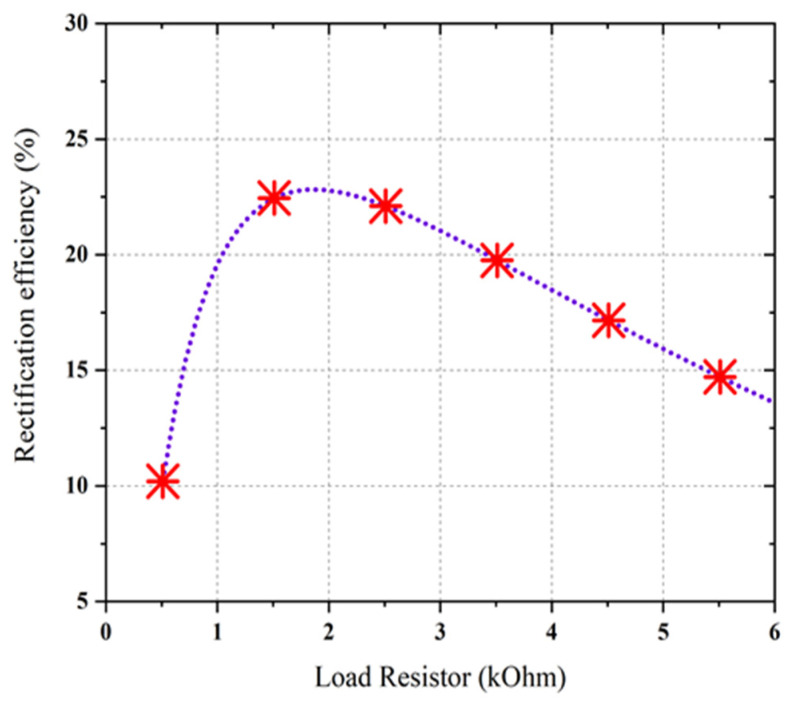
The measured overall rectification efficiency as a function of different load resistance with an error (cross signs).

**Table 1 sensors-21-07838-t001:** Various performances comparison between new design and previous designed related works.

Ref. No	Band of Freq.	Antenna Size (mm)	No. of Port	Frequencies (GHz)	Antenna Gain (dBi)	Max. Effi. at −20 dBm (%)
[6]	dual	190 × 100	1	1.84, 2.14	10.9, 13.3	34, 29
[41]	triple	160 × 160	1	2, 2.5, 3.5	7, 5.5, 9.2	20, 7, 5
[42]	hexa	160 × 160	1	0.55, 0.75, 0.90, 1.85, 2.15, 2.45	2.5, 3.1, 3.6, 5.0, 5.0, 4.5	25, 20, 25,15, 9, 5
[43]	triple	175 × 200	2	0.94, 1.84, 2.14	8.15, 7.15, 8.15	27.3, 20, 14
[44]	single	226 × 337	2		15.5 for Ports 1–2	N/A.
[45]	single	150 × 150	8	2.4	Appx. 5 for Ports 1–8	28.6
[46]	dual	240 × 240	16	0.94, 1.84	Appx. 3.6 for Ports 1–4 Appx. 3.8 for Ports 5–16	38.1, 34
[47]	triple	200 × 200	16	1.84, 2.14, 2.45	9, 11, 11	25.3, 27.9, 19.3
[48]	triple	88.5 × 40	1	0.90, 1.80, 2.10	N/A	31.2
[49]	triple	145 × 145	1	1.8, 2.15, 2.45	4.33,4.22,3.88	67
[50]	quad	245.1 × 150		0.84, 1.86, 2.10, 2.45	N/A	30, 22, 33, 16.5 @ −25, −5
New	quad	160 × 160	2	0.850, 1.81, 2.18, 2.40	3.95, 4.45, 4.42, 4.82	48

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
