# Peer review of "Quad-Band Rectenna for Ambient Radio Frequency (RF) Energy Harvesting"

_sensors, 2021, doi:10.3390/s21237838_

Round 1

Reviewer 1 Report

In this paper the authors presents a RF energy harvesting multi-band rectenna that works well from 1.8 to 2.5 GHz. The paper is well written and the methods are well explained but the manuscript don't present anything new in relation to the state of the art. All present here is well known and published in several papers in the literature. 

There are no scientific contributions to improve the state of the art. 
moreover all presented results is with -20 dBm of input power. what's happens if the power is bigger? And if it is smaller? -20 dBm is a not realistic value for ambient signal levels. This is true if the distance is very small (less than 1 m) 

Author Response

All issues are addressed in the uploaded file. 

Reviewer 2 Report

This manuscript presents a quad-band rectenna for ambient RF energy harvesting. I have following comments.

  1. What is the polarization of the proposed cross dipole? In line 186-187, it is mentioned that “form the right-hand circular polarization radiation field and left-hand circular polarization radiation field in front and backside of the antenna”. However, in line 223, it is said that “it has the linear directional polarization features”. The authors should clarify the polarizations.
  2. For the measured realized gain in Fig. 4, the reviewer wonder what is the radiation efficiency of the proposed quad-band antenna. Please clarify this.
  3. In Fig. 12, what is the input RF power for measuring S11 of the rectifier? Is such input RF power consistent with the low input RF power in ambient RF energy harvesting?
  4. In Fig. 14, can the authors provide more details about how to measure the RF-to-dc efficiency with the dual-tone, three-tone, and four-tone signals? How to generate these multi-tone signal?
  5. In Fig. 16, for the measurement with artificial RF energy source, is the antenna directly connected to the rectifier? Or the rectifier is only connected to the signal generator? Please make it clear.
  6. In Fig .19, the port 1 and port 2 of the rectifiers are separately connected to the single port of antenna. However, in practice, how to simultaneously harvesting energy with port 1 and port 2? It seems like two antennas are needed to separately connect to port 1 and port 2 of the rectifier. Please solve this problem.
  7. It is suggest to compare more works on multiband rectenna such as 1109/IMWS-BIO.2014.7032426, 10.1109/TIE.2020.3009586,  10.1109/LMWC.2020.3029869
  8. Please carefully check the reference since there are many repeated references, such as [4] and [5], [6] and [11], [20] and [21], [23] and [24], [43] and [44].

Author Response

The response letter is uploaded below.

Reviewer 3 Report

The paper presents a dual-port quad-band RF energy harvester with large bandwidth and small size using triple branches and a new impedance matching technique. A new rectifier circuit is designed to improve the power sensitivity by reducing the consumption of RF power. The numerical simulations are performed to study the S-parameters, gain, surface current distribution at different frequency bands of interest, and radiation patterns and compared with experimental results. The prototype of the proposed energy harvesting antenna was manufactured and tested at the frequency range of interest and ambient environment. The results show that the efficiency could be up to 48%, which outperforms the results in literature.

The presented research is interesting and the results are exciting. However, the following concerns need to be addressed before it could be accepted for publishing.

  • It’s not clear how the S-parameters of the proposed antenna in Fig. were measured and simulated? It’s suspicious that the simulated results of the proposed antenna almost perfectly agree with the measurements. Clear descriptions of the simulation and measuring might make the results more solid and convincing.
  • Again, it’s not clear how the results in Fig. 4 were derived. On page 6, lines 210-212, it stated the difference between the simulation results and measurements, however, didn’t analyze the reason.
  • Most of the results presented in the paper look very good and well agreement between numerical simulations and experimental results were observed. However, the simulation conditions and parameters were not given and the test setup was not well elaborated either. Only Fig 16-19 show some of the experimental setup.

Author Response

pls, check the attachment. 

Round 2

Reviewer 1 Report

My opinion about the manuscript is the same. The paper is more one about Energy Harvesting. There are nothing new that we can point. The paper don't contribute to improve the state of the art. Why the paper must be published? Please said on new point in this paper that don´t exist in the state of the art. 

The authors don't explain the impact of input power variation and the circuit is design for an input power of -20 dBm.  If the power is not this value the system have a bad performance. 

Author Response

We are grateful that the reviewer observes our manuscript improvements. We are also sorry that reviewers found a lack of scientific insight on the contribution given about some comments. However, we have thoroughly revised and updated our previous manuscript based on the reviewer's recommendation.

pls check the attached file as a response.

Reviewer 2 Report

The authors have addressed all my concerns. I recommend accept.

Author Response

thank you.